# Transvaginal Sonographic Evaluation of Cesarean Section Scar Niche in Pregnancy: A Prospective Longitudinal Study

**DOI:** 10.3390/medicina57101091

**Published:** 2021-10-12

**Authors:** Egle Savukyne, Egle Machtejeviene, Saulius Paskauskas, Gitana Ramoniene, Ruta Jolanta Nadisauskiene

**Affiliations:** 1Department of Obstetrics and Gynaecology, Hospital of Lithuanian University of Health Sciences, Kauno Klinikos, Eiveniu st. 2, 50161 Kaunas, Lithuania; egle.machtejeviene@lsmuni.lt (E.M.); saulius.paskauskas@lsmuni.lt (S.P.); gitana.ramoniene@lsmuni.lt (G.R.); ruta.nadisauskiene@lsmu.lt (R.J.N.); 2Department of Obstetrics and Gynaecology, Medical Academy, Lithuanian University of Health Sciences, A.Mickevicius st. 7, 44307 Kaunas, Lithuania

**Keywords:** Cesarean scar, Cesarean scar niche, transvaginal ultrasonography

## Abstract

*Background and Objectives*: To investigate the prevalence of a Cesarean section (CS) scar niche during pregnancy, assessed by transvaginal ultrasound imaging, and to relate scar measurements, demographic and obstetric variables to the niche evolution and final pregnancy outcome. *Materials and Methods:* In this prospective observational study, we used transvaginal sonography to examine the uterine scars of 122 women at 11^+0^–13^+6^, 18^+0^–20^+6^ and 32^+0^–35^+6^ weeks of gestation. A scar was defined as visible on pregnant status when the area of hypoechogenic myometrial discontinuity of the lower uterine segment was identified. The CS scar niche (“defect”) was defined as an indentation at the site of the CS scar with a depth of at least 2 mm in the sagittal plane. We measured the hypoechogenic part of the CS niche in two dimensions, as myometrial thickness adjacent to the niche and the residual myometrial thickness (RMT). In the second and third trimesters of pregnancy, the full lower uterine segment (LUS) thickness and the myometrial layer thickness were measured at the thinnest part of the scar area. Two independent examiners measured CS scars in a non-selected subset of patients (*n* = 24). Descriptive analysis was used to assess scar visibility, and the intraclass correlation coefficient (ICC) was calculated to show the strength of absolute agreement between two examiners for scar measurements. Factors associated with the CS scar niche, including maternal age, BMI, smoking status, previous vaginal delivery, obstetrics complications and a history of previous uterine curettage, were investigated. Clinical information about pregnancy outcomes and complications was obtained from the hospital’s electronic medical database. *Results:* The scar was visible in 77.9% of the women. Among those with a visible CS scar, the incidence of a CS scar niche was 51.6%. The intra- and interobserver agreement for CS scar niche measurements was excellent (ICC 0.98 and 0.89, respectively). Comparing subgroups of women in terms of CS scar niche (*n* = 49) and non-niche (*n* = 73), there was no statistically significant correlation between maternal age (*p* = 0.486), BMI (*p* = 0.529), gestational diabetes (*p* = 1.000), smoking status (*p* = 0.662), previous vaginal delivery after CS (*p* = 1.000) and niche development. Uterine scar niches were seen in 56.3% (18/48) of the women who had undergone uterine curettage, compared with 34.4% (31/74) without uterine curettage (*p* = 0.045). We observed an absence of correlation between the uterine scar niche at the first trimester of pregnancy and mode of delivery (*p* = 0.337). Two cases (4.7%) of uterine scar dehiscence were confirmed following a trial of vaginal delivery. *Conclusions:* Based on ultrasonography examination, the CS scar niche remained visible in half of the cases with a visible CS scar at the first trimester of pregnancy and could be reproducibly measured by a transvaginal scan. Previous uterine curettage was associated with an increased risk for uterine niche formation in a subsequent pregnancy. Uterine scar dehiscence might be potentially related to the CS scar niche.

## 1. Introduction

Defects of the uterine scar seem to be a rapidly increasing problem caused by increasing Cesarean delivery rates worldwide. In recent decades, there has been an increasing number of studies that describe Cesarean section (CS) scars, but it is not known how the CS scar niche is associated with an increased risk of uterine dehiscence or rupture in labor. Ultrasonographic evaluation of the uterine scar has become an important element of obstetric and gynecologic practice, especially in further pregnancies. However, there is still limited evidence relating CS scar niche to pregnancy outcome, yet the visibility and measurement of a CS scar on ultrasound examination may be clinically relevant. Many studies describing CS scars using ultrasonography have been carried out on non-pregnant and pregnant subjects [1,2,3,4,5]. A CS scar niche is described as a triangular, hypoechoic area defect at the site of a previous CS. It comprises two components: the hypoechoic part and the tissue contained within the residual myometrium [1,5]. A large niche is defined as an incision of a depth of at least 50% or 80% of the anterior myometrium, or the remaining myometrial thickness ≤ 2.2 mm evaluated by TVS [6]. Previous studies proposed a standardized approach for assessing the hypoechoic component of the scar as the RMT measurement [5]. The reported prevalence of CS niche varies within −24–70% using transvaginal scans (TVS) [3,5,7]. There is a lack of evidence about potential risk factors of the CS scar niche. According to the literature, gestational diabetes, previous Cesarean delivery and advanced body mass index are independent risk factors for niche development [8]. Few methods have been used to correlate measurements of the lower uterine segment (LUS) on pregnant status with the risk of uterine dehiscence or rupture [9,10,11]. The LUS was measured by transabdominal ultrasound, and in other studies, the muscular layer was measured by TVS [12]. Unfortunately, no cut-off values have been tested. It is not known whether the CS niche influences the type of delivery and pregnancy outcomes. Previous studies showed that TVS could reproducibly measure the CS scar niche, RMT and LUS thickness, with good agreement between two observers [13]. Nevertheless, there is a lack of studies describing CS scars during pregnancy.

The study aimed to determine how accurately CS scars can be detected by transvaginal ultrasonography and estimate the CS scar niche’s prevalence. Moreover, we tested the reproducibility of the CS scar measurements throughout pregnancy. We investigated some of the obstetric and demographic variables associated with niche development in a subsequent pregnancy and related CS scar niche and pregnancy outcomes.

## 2. Materials and Methods

The study was carried out at the Hospital of the Lithuanian University of Health Sciences Kaunas Clinics, Kaunas, Lithuania, approved by the Kaunas regional bioethics committee (protocol No. BE-10–15). The trial registration date was March 18, 2019 (Australian New Zealand Clinical Trials Registry, Registration No. ACRN12619000435189). Informed consent was obtained from all participants after the nature of the ultrasound examination had been fully explained. Between March 2019 and October 2020, 140 patients were enrolled in the study. We included women with singleton pregnancies after at least one previous low-transverse CS. Women with twin gestations (*n* = 3) were excluded from the study, as after classical CS (*n* = 2). Five patients withdrew from the study after the first trimester scan, and eight were lost to follow-up, so the final analysis consisted of 122 women (Figure 1).

The CS scar’s ultrasound measurements were performed with a Voluson E8 Expert (GE Healthcare Ultrasound Korea, Ltd) ultrasound system equipped with a 5–9 MHz transvaginal probe. 

The first researcher performed all ultrasound examinations in the lithotomy position, with an empty bladder at the first trimester scan and filled bladder at the second and third trimesters. All participating patients underwent TVS at 11–14, 18–21 and 32–36 weeks of gestation, coinciding with routine first trimester screening, fetal anomaly screening and growth assessment scans. Ultrasound images of a CS scar were classified subjectively as (1) a non-visible scar, (2) visible scar without niche, (3) CS scar niche, when the depth of the hypoechoic part was 2 mm or more, as shown in Figure 2. When a CS scar was not visible or was without a scar niche, the myometrial thickness of the isthmus uteri at the level of the internal cervical os was measured. When a CS scar was visible, measurements were taken in the sagittal plane, the length (widest gap along the cervical canal), the depth of the hypoechoic scar part (the vertical distance between the base and apex of the visible defect) and RMT over the niche (defined as the distance from the bottom of the niche to the uterovesical fold). The ratio (expressed as a percentage) between the thickness of the remaining myometrium over the niche and the thickness of the myometrium adjacent to and fundal to the defect was calculated. The scar measurement technique is presented in Figure 3. LUS and myometrial thickness at the area likely containing a CS scar were assessed in the second and third trimesters of pregnancy, as previously described [14]. All representative images were stored on the local hospital image storage system (DICOM). Data were recorded prospectively on an SPSS spreadsheet. The second part of the study focused on the inter- and intraobserver variability for measuring CS scars. Overall, twenty-four consecutive cases were assessed by real-time scanning to evaluate the visibility of CS and measurements of CS scar niche as LUS and myometrial thickness by two independent investigators at every trimester of pregnancy. At each examination episode, the first researcher assessed the lower uterine segment in the sagittal plane to identify the CS scar area. The measurements then were taken: scar length, depth, RMT and myometrial thickness adjacent to the niche. The second researcher took over the examination of the same patient. He was blinded to the findings of the first operator and repeated all the measurements twice. For intraobserver variability, the first researcher carried out the measurements for each scar dimension twice and results were recorded in a database. Patients management and delivery of pregnancy were carried out according to the hospital’s policy. Obstetricians were blinded to the results of the CS scar ultrasound evaluation. All data about pregnancy outcomes and complications were retrieved from hospital records after delivery.

Statistical analyses were performed using SPSS Statistics v 27.0 (IBM Corp., Armonk, NY, USA). The median and 25th–75th percentile for the length and depth of the scar niche as RMT were calculated in the first trimester of pregnancy, and LUS and myometrial thickness in the second and third trimesters. The statistical significance of the difference in categorical data was determined using the Chi-square test or Fisher’s exact test. Continuous variables were compared between groups (after one CS versus after two or more CS; CS scar niche group versus non-niche group) using the Mann–Whitney U-test. *p* < 0.05 was considered statistically significant. To determine systematic bias in uterine scar measurements between two researchers, the mean of differences and its standard error (SEM) were calculated. The limits of agreement between two investigators were calculated for each measurement as the mean ± (1.96 × SD). For inter- and intraobserver agreement, the intraclass correlation coefficient (ICC) was calculated. High absolute agreement corresponds to a high ICC (close to 1), with values > 0.75 indicative of a test with good agreement [15].

## 3. Results

Overall, 122 participants were included to assess CS scar visibility. The CS scar was visible in 95/122 (77.9%) cases, as the uterine scar niche was found in 49/122 (40.2%). Of those with visible CS scars, half of the women had a CS niche 49/95 (51.6%). All of the CS scar niches were triangular in shape (Figure 2) and 36/49 (73.4%) were large ones (an incision of a depth of at least 50% of the anterior myometrium). Only two patients (1.6%) displayed a retroverted uterus at the first trimester of pregnancy, both with a non-visible uterine scar after one previous CS. There was 100% agreement on scar niche visibility at the 11–13-week scan between the two researchers. Demographic background data are shown in Table 1.

CS scar niche size and dimensions of a low uterine segment as myometrial thinning in each trimester of pregnancy are shown in Table 2. CS scar niches were seen in the first trimester of pregnancy only. Systematic bias between the two observers was assessed. We found that there were no systematic biases between researchers during measurements in all three trimesters. The means of differences between the two observers and the limits of agreement are shown in Table 3. For CS scar niche measurements, the limits of agreement were narrow in all trimesters of pregnancy.

For intraobserver comparisons, we calculated that the difference of ≤1 mm in the CS scar niche measurements obtained by two observers was acceptable [14]. The difference was ≤1 mm in 92% of cases for the scar measurements, as RMT over the scar niche was 100%. For myometrial thickness at internal cervical os, the difference was ≤1 mm in 85% of cases. Intraobserver agreement was perfect for all scar measurements during all trimesters of pregnancy, with ICC between 0.75 and 0.97 for all scar measurements (Table 4). Interobserver agreement for CS scar niche length, depth and RMT between two different observers in the first trimester within the acceptable range for ≤1 mm was 85.7%. Overall, ICC was >0.8 for all measurements across all trimesters of pregnancy and >0.7 for CS scar niche length (Table 4).

Most included participants had undergone one CS 94/122 (77.4%), 26 underwent two CS (21.3%) and only two cases were after three CS (1.6%). CS scar niche was visible as often in women after one previous CS as in those after two and more previous CS (83.7% (41/94) vs. 16.3% (8/28), *p* = 0.228). Nevertheless, non-visible CS scars were more often seen in patients after one previous CS compared with those after two and more CS (63% (17/94) vs. 37% (10/28), *p* = 0.049). The LUS and myometrial layer in the second trimester were thinner after two or more previous CS than after one CS (*p* = 0.012). Nevertheless, there were no differences in the third trimester of pregnancy (*p* = 0.503) (Table 5).

Statistical analysis was carried out to investigate the influence of maternal and obstetric history for CS scar niche formation in a subsequent pregnancy. We found that previous uterine curettage significantly influenced uterine scar niche development (*p* = 0.049). In contrast, no significant association was found between CS niche and maternal age, BMI, gestational diabetes, smoking status and previous vaginal birth (Table 6).

Of the 63 women who underwent a trial of labor after one previous CS, 41 (65.0%) had a successful vaginal delivery. Vacuum extraction was performed in one case (1.5%) among these women. Of 27 (42.8%) women with labor induction after one previous CS, 15 (55.5%) had a vaginal delivery. Emergency repeat CS after trial of labor was performed in 23 (36.5%) women, 10 due to non-reassuring fetal status and 13 due to arrest of labor. In the patient group with visible CS scars at the first trimester of pregnancy, seven delivered through natural pathways, as did the same number of women in the group with non-visible CS scars (55.7% vs. 58.3%, *p* = 1.000). In the patient group with CS scar niches (*n* = 49) for comparison with the non-niche group (*n* = 46), there were no statistical differences in the type of delivery. A total of 19 women had successful trials of labor in the niche group and 22 in the non-niche group (38.7% vs. 47.8%, *p* = 0.802). Fifteen underwent elective repeat Cesarean delivery for various clinical reasons in the niche group, in comparison with 33 women in the non-niche group (31.9% vs. 44.6% *p* = 0.337). Thirteen women required intrapartum emergency CS because of failed trials of labor in the niche group, versus 19 women in the non-niche group (40.6% vs. 46.3% *p* = 0.802). The mean gestational age of the study population at delivery was 38.8 ± 2.37 weeks, with a neonatal weight of 3473.7 ± 598.0 g. The median neonatal weight did not differ between the groups according to the CS scar niche and the Apgar score. Median birth weight in the CS scar niche group was 3589.0 (IQR 3183.7–3922.0) g, compared with 3515.0 (IQR 3260.0-3685.0) g in the non-niche women’s group (*p* = 0.340). After 5 min in both groups, the median Apgar score was 10.0 (IQR 9.0–10.0) (*p* = 0.951). There were two (4.3%) cases with a neonatal Apgar score less than 7 after one minute in the CS scar niche group, compared with a single case (1.4%) in the non-niche women’s group. Two women (4.3%) had uterine dehiscence confirmed following a trial of vaginal delivery, and both had CS niches during the first trimester. No uterine ruptures occurred in the study population.

## 4. Discussion

Our study found that previous uterine curettage was associated with uterine scar niche development in a subsequent pregnancy. We also report here that CS scars can be accurately detected with a high level of agreement between observers in all trimesters of pregnancy by TVS.

The study’s strengths are its prospective design and the fact that the reproducibility of the CS scar measurements was determined in all three trimesters of pregnancy. All TVS scans were performed by the first author and a fraction of the scans by two observers with experience in the assessment of CS scars. All the scans were performed according to a standardized procedure in the sagittal plane [5,8,16]. Our study showed a CS scar visibility rate of 77.9%; in previous studies, CS scar visibility with ultrasound ranged between 7 and 89% [5,13]. The prevalence of large niches in our study population was 73.4%, compared with 10–42% in the previous two studies [2,17]. This discrepancy could be explained by population differences, as previous studies’ measurements of the CS scar “defect” were taken in non-pregnant status [2,3,4,18]. They suggested that the CS scar niche is often seen in the retroverted uterus [2]. In our prospective study, we had only two cases of a retroverted uterus in the first trimester, both patients with non-visible CS scars. In a previous study by Naji et al., they found that, in 36 cases, a CS scar was not visible to operators when the uterus was retroverted [19]. Our results of inter- and intraobserver variability analysis for scar measurements during the first, second and third trimesters were excellent, and this is a better result in comparison with other studies reporting good agreement between two observers regarding uterine scar measurements at the first trimester and moderate for the second and third trimesters [13,14]. This may be caused by performing the scar assessment offline on the stored images in previous studies. In our study, two operators used real-time scanning of the CS scar one after another. However, there was a degree of systematic bias because our analysis was performed within routine ultrasound screening at the first and second trimesters, lacking time for additional TVS scans. The limitation of the study is the small number of patients enrolled in the reproducibility subgroup.

In the present study, 22.1% of women had a non-visible CS scar at the first trimester of pregnancy and these were not clearly seen in the second or third trimesters of pregnancy. This may be due to ultrasound’s poor ability to detect pregnancy-related anatomical changes in the scar area [16]. Therefore, in patients with non-visible CS scars, we measured the thinnest part of the myometrium at the internal cervical os, as was done in previous studies [17]. Jordan et al. [20] published an international consensus statement about the CS scar niche in non-pregnant status. Nonetheless, there is no consensus on the CS scar niche measurements in pregnant status to the best of our knowledge. To date, the exact causes of niche formation remain unresolved, but there have been many reports about the risk factors for CS scar niche development [2,8,21,22,23]. It seems that it depends on the various patient, pregnancy and previous surgery factors—the number of previous CS, the site of hysterotomy, the suturing technique and maternal conditions such as gestational diabetes, smoking or BMI [8,18,21]. However, we did not find a relationship between previous CS deliveries, patient’s BMI, smoking status, gestational diabetes and uterine scar niche development. This discrepancy with earlier studies may be caused by the small number of patients with multiple CS scars in our study population. Nevertheless, a relationship between multiple CS and scar niche has been reported previously [2,8,22,24]. It is known that a uterine scar after CS negatively influences new scar healing because of repeated trauma to the isthmic wall and reduced vascular perfusion in the surrounding uterine scar [2,25]. We have shown here for the first time that previous uterine surgery as curettage raises the risk of CS scar niche in a subsequent pregnancy. According to the above literature and our results, we believe that there are similarities between multiple CS scars in the lower uterine segment and previous uterine curettage as a risk factor for incomplete healing of the scar tissue or myometrial damage. An association between large scar defects in non-pregnant women and uterine rupture/dehiscence of scars in a subsequent delivery has been suggested [26]. It was thought that first-trimester CS scar assessment would be a promising tool to identify high-risk patients for uterine dehiscence/rupture or placenta accreta spectrum disorders at the beginning of the pregnancy [4,27]. However, previous investigators could not conclude that a single evaluation of the CS scar in the first trimester could be used to predict uterine rupture/dehiscence too [4,15,19,28]. Kim Paquette et al. previously assessed uterine scars by transvaginal ultrasound in the first trimester and then in the third trimester of pregnancy in 166 women and concluded that evaluation of the CS scar in the first trimester could not be used as a predictor of uterine rupture in a subsequent pregnancy [28]. Our findings suggest the same—there is no such association. The main limitation of our study is the small patient number. Many women were referrals and previous CS had been performed by different obstetricians in different institutions; however, in Lithuania, we have a uniform technique for Cesarean delivery that involves the single-layer suturing of the uterine incision.

## 5. Conclusions

Our study demonstrates that the CS scar niche is a myometrial defect, which can be detected by TVS at the first trimester of pregnancy and is associated with previous uterine curettage, but is not necessarily associated with undesirable pregnancy outcomes. The study results cannot provide recommendations regarding routine ultrasound examinations of CS scars in pregnant women in order to appropriately manage subsequent deliveries. However, we suggest that women in whom a uterine scar niche is detected at first-trimester scan are likely to be a high-risk group for uterine dehiscence or rupture during delivery. Women should avoid CS without medical indications and multiple abortions with uterine curettage. Nevertheless, more prospective high-quality studies are needed to establish the clinical significance of the CS scar niche and to define guidelines for the possible prevention of the CS scar niche in a subsequent pregnancy.

## Figures and Tables

**Figure 1 medicina-57-01091-f001:**
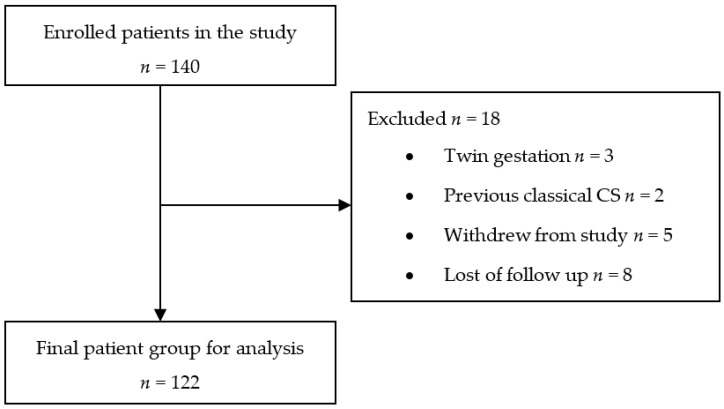
Flowchart of the patients enrolled in the study.

**Figure 2 medicina-57-01091-f002:**
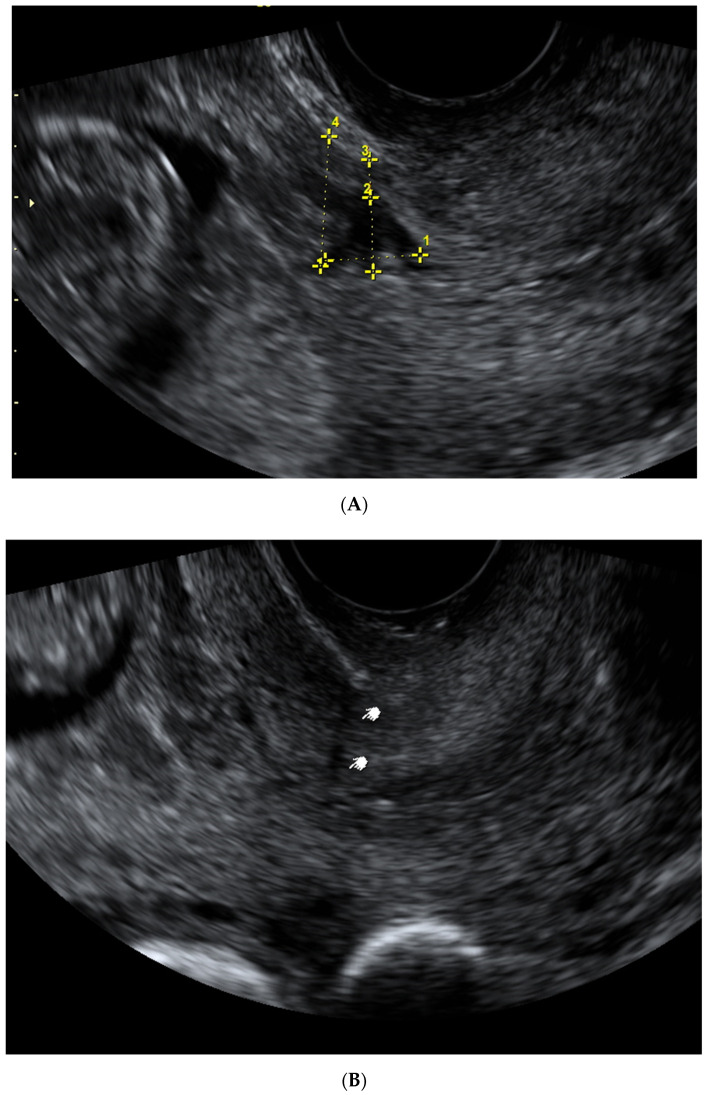
First-trimester ultrasound images of a CS scar niche (**A**), a visible CS scar (**B**) and a non-visible CS scar (**C**).

**Figure 3 medicina-57-01091-f003:**
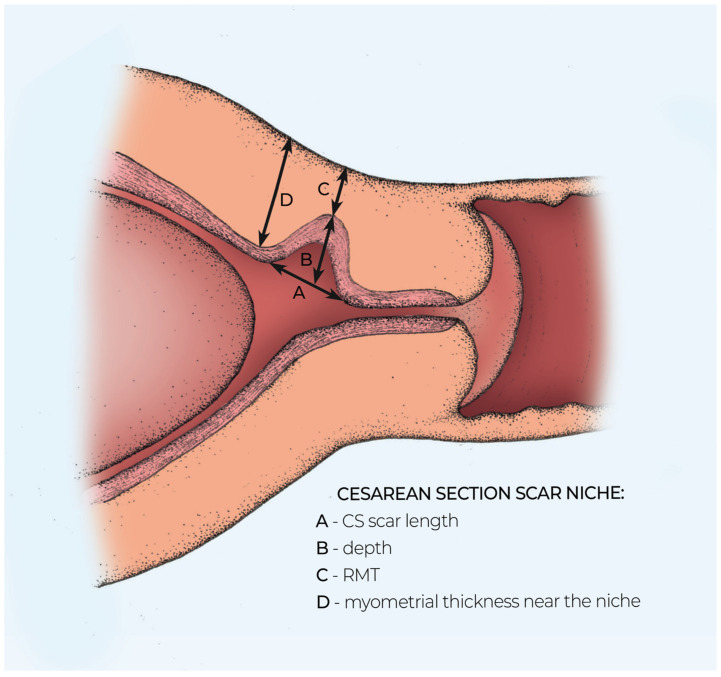
Schematic diagram showing Cesarean scar niche dimensions in the sagittal plane. A, niche length—the length of a hypoechoic part of the niche; B, depth of the hypoechogenic part of the niche; C, residual myometrial thickness; D, the thickness of the myometrium adjacent to the niche.

**Table 1 medicina-57-01091-t001:** Demographic and obstetric history of the study population (*n* = 122).

Characteristic	Median (IQR) or *n* (%)
Maternal age (years)	34 (29–34)
BMI (kg/m^2^)	25 (21–28)
Gestational diabetes	15 (12.3)
Hypertension	12 (9.8)
Smoking during pregnancy	16 (13.1)
Previous uterine curettage	32 (26.2)
Previous VBAC	8 (6.6)
Previous postpartum infection	5 (4.1)

BMI, body mass index at first trimester; VBAC, vaginal birth after Cesarean section; IQR, interquartile range.

**Table 2 medicina-57-01091-t002:** Size of scar niche and dimensions of LUS and myometrial thickness.

Scar Characteristic (mm)	First Trimester Median (IQR)	Second Trimester Median (IQR)	Third Trimester Median (IQR)
CS scar niche length	5.0 (3.9–7.0)	-	-
CS scar niche depth	6.9 (4.8–9.7)	-	-
RMT	4.7 (3.3–5,9)	-	-
Myometrial thickness in the isthmus uteri	12.7 (10.8–14.5)	-	-
LUS thickness	-	6.85 (5.2–9.1)	4.0 (0.9–5.5)
Myometrial thickness	-	3.85 (2.5–5.3)	2.1 (1.7–2.7)

CS, Cesarean section; RMT, residual myometrial thickness; LUS, low uterine full segment thickness; IQR, interquartile range.

**Table 3 medicina-57-01091-t003:** Mean (±SEM) of differences between and 95% limits of agreement (LoA) for CS scar niche and low uterine segment measurements by two researchers across all three trimesters (*n* = 24).

Scar Characteristic (mm)	First Trimester Mean ± SEM of Difference (95% LoA)	Second Trimester Mean ± SEM of Difference (95% LoA)	Third Trimester Mean ± SEM of Difference (95% LoA)
CS scar niche length	−0.300 ± 0.467 (+3.723 to 3.123)	-	-
CS scar depth	−0014 ± 0.211 (−1.565 to 1.536)	-	-
RMT	−0.114 ± 0.140 (−1.142 to 0.914)	-	-
Myometrial thickness in the isthmus uteri	−0.133 ± 0.169 (−1.761 to 1.494)	-	-
LUS thickness	-	−0.104 ± 0.169 (−1.725 to 1.517)	0.021 ± 0.077 (−0.717 to 0.759)
Myometrial thickness	-	−0.083 ± 0.216 (−2.158 to 1.999)	0.004 ± 0.036 (−0.340 to 0.348)

CS, Cesarean section; RMT, residual myometrial thickness; LUS, low uterine segment.

**Table 4 medicina-57-01091-t004:** Intraobserver (obtained by one researcher, with a short interval between measurements) and interobserver agreement for CS scar niche measurements and RMT in the first trimester, LUS and myometrial thickness in the second and third trimesters (*n* = 24).

	Percentage of Difference ≤ 1 mm	Intraclass Correlation Coefficient (95% CI)
**Intraobserver Agreement**		
CS scar niche length	92.9	0.984 (0.95–0.995)
CS scar niche depth	92.9	0.989 (0.968–0.997)
RMT	100	0.991 (0.971–0.991)
Myometrial thickness at internal os	83.3	0.96 (0.91–0.983)
LUS second trimester	95.8	0.992 (0.982–0.997)
Myometrial thickness second trimester	95.8	0.979 (0.953–0.991)
LUS third trimester	91.7	0.914 (0.811–0.962)
Myometrial thickness third trimester	91.6	0.906 (0.796–0.958)
	**Percentage of difference ≤ 1 mm**	**Intraclass correlation coefficient (95% CI)**
**Interobserver Agreement**		
CS scar niche length	85.7	0.758 (0.399–0.915)
CS scar niche depth	85.7	0.972 (0.916–0.991)
RMT	85.7	0.969 (0.906–0.995)
Myometrial thickness at internal os	87.5	0.966 (0.924–0.985)
LUS second trimester	87.5	0.967 (0.925–0.986)
Myometrial thickness second trimester	91.7	0.854 (0.692–0.934)
LUS third trimester	100	0.969 (0.929–0.986)
Myometrial thickness third trimester	100	0.918 (0.824–0.964)

CS, Cesarean section; RMT, residual myometrial thickness; LUS, low uterine segment.

**Table 5 medicina-57-01091-t005:** Ultrasound findings in women after one (*n* = 94) or two or more (*n* = 28) previous CS.

Finding	One CS Median (IQR) or *n* (%)	Two and More CS Median (IQR) or *n* (%)	*p* Value
Visible CS scar	77 (81.1)	18 (18.9)	0.049
Non-visible CS scar	17 (63)	10 (37)	0.049
CS scar niche	41 (43.6)	8 (28.5)	0.228
RMT	4.8 (3.6–6.6)	3.4 (1.9–5.3)	0.09
RMT ≤ 2 mm	3 (60.0)	2 (40.0)	0.323
Ratio (%)	35.2 (25.7–51.34)	31.2 (24.9–47.7)	0.532
≤50	30 (73.2)	6 (75)	0.645
LUS thickness in second trimester	7.4 (5.4–9.5)	6.0 (4.6–7.9)	0.012
Myometrial thickness in second trimester	4.0 (2.77–5.4)	2.5 (2.0–5.3)	0.022
LUS thickness in third trimester	4.0 (2.7–5.4)	4.1 (2.8–5.1)	0.503
Myometrial thickness in third trimester	2.1 (1.6–2.8)	2.3 (1.7–2.6)	0.97

CS, Cesarean section; RMT, residual myometrial thickness; LUS, low uterine segment. Ratio between the RMT and the thickness of the myometrium adjacent to the defect.

**Table 6 medicina-57-01091-t006:** Influence of demographic and obstetric variables on CS niche development in a subsequent pregnancy.

Parameter	CS Scar Niche Median (IQR) or *n* (%)	Without CS Scar Niche Median (IQR) or *n* (%)	*p* Value
Age (years)	34 (27.9–36.0)	35 (26.0–37.0)	0.486
BMI (kg/m^2^)	24.9 (21.6–28.5)	25.5 (21.9–28.5)	0.529
Gestational diabetes	6 (12.2)	9 (12.3)	1
Smoker	5 (10.4)	11 (14.9)	0.662
Previous VBAC	3 (6.25)	5 (6.8)	1
Uterine curettage	18 (56.3)	31 (34.4)	0.049

BMI, body mass index; VBAC, previous vaginal delivery after Cesarean section.

## Data Availability

The data presented in this study are available on request from the corresponding author.

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
