# Peer review of "Transvaginal Sonographic Evaluation of Cesarean Section Scar Niche in Pregnancy: A Prospective Longitudinal Study"

_medicina, 2021, doi:10.3390/medicina57101091_

Round 1

Reviewer 1 Report

interesting topic

sufficient scientific and litterature information

Author Response

Dear Reviewer,

Thank you for your reply. We are happy you found the topic is interesting and of good quality.

Reviewer 2 Report

It is a very interesting article regarding sonographic evaluation of cesarean section scar status during pregnancy. There is a detailed description of the technique used for scar evaluation, as well as identification of scar defects. There is an emphatic presentation of ultrasonographic findings and their importance in pregnancy outcome. Moreover, authors investigated predisposing factors for scar defect formation. Furthermore, they discuss their findings and compare them with previous studies.

Author Response

Dear Reviewer,

Many thanks for your reply. We are grateful for your comments and assessment.

Reviewer 3 Report

A well-designed and informative publication. The only missing information is the way the uterus is being opened and sutured in your hospital, one or two layers, as you write in your discussion (line 276-277) that the niche depends among others on the site of the uterine incision and the technique used for suturing. This is an important information, as the literature is controversial concerning the optiman site to open the uterus (above or below the bladder plica) and the way to suture the uterus (one or two layers).

Author Response

Dear Reviewer,

Thank you very much for your comments and suggestions. We found them very useful to improve a manuscript.

Reply to a reviewer:

In Lithuania, we have the same cesarean section operation technic as in our clinic - we close the uterus in one layer, and opening the uterus above plica vesicouterine mainly.  Unfortunately, in our study, we do not evaluate uterine suture technic, as in our country we have the same technic mainly as we mentioned in our manuscript (last sentence in a discussion).

"but in Lithuania we have uniform technic for cesarean delivery mostly"